# Missing checkerboards? An absence of competitive signal in *Alnus*-associated ectomycorrhizal fungal communities

Peter Kennedy[1,2], Nhu Nguyen[1], Hannah Cohen[2] and Kabir Peay[3]

[1] Department of Plant Biology, University of Minnesota, St. Paul, MN, USA
[2] Department of Biology, Lewis & Clark College, Portland, OR, USA
[3] Department of Biology, Stanford University, Palo Alto, CA, USA

## ABSTRACT

A number of recent studies suggest that interspecific competition plays a key role in determining the structure of ectomycorrhizal (ECM) fungal communities. Despite this growing consensus, there has been limited study of ECM fungal community dynamics in abiotically stressful environments, which are often dominated by positive rather than antagonistic interactions. In this study, we examined the ECM fungal communities associated with the host genus *Alnus*, which live in soils high in both nitrate and acidity. The nature of ECM fungal species interactions (i.e., antagonistic, neutral, or positive) was assessed using taxon co-occurrence and DNA sequence abundance correlational analyses. ECM fungal communities were sampled from root tips or mesh in-growth bags in three monodominant *A. rubra* plots at a site in Oregon, USA and identified using Illumina-based amplification of the ITS1 gene region. We found a total of 175 ECM fungal taxa; 16 of which were closely related to known *Alnus*-associated ECM fungi. Contrary to previous studies of ECM fungal communities, taxon co-occurrence analyses on both the total and *Alnus*-associated ECM datasets indicated that the ECM fungal communities in this system were not structured by interspecific competition. Instead, the co-occurrence patterns were consistent with either random assembly or significant positive interactions. Pair-wise correlational analyses were also more consistent with neutral or positive interactions. Taken together, our results suggest that interspecific competition does not appear to determine the structure of all ECM fungal communities and that abiotic conditions may be important in determining the specific type of interaction occurring among ECM fungi.

Corresponding author
Peter Kennedy, kennedyp@umn.edu

## INTRODUCTION

A common way to assess the role of interspecific competition or facilitation in determining community structure is experimental manipulation involving the removal of neighboring individuals. This approach has been widely used in ecological studies examining biotic determinants of plant and animal communities (*Connell, 1983*; *Schoener, 1983*), but carrying out similar manipulations in field-based studies of soil microbial communities

is less feasible due to the inability to selectively manipulate species-level neighborhood composition. A widely proposed alternative is to look at species distribution patterns, with *Diamond*'s (*1975*) study of bird distributions in the New Guinea archipelago being one of most well-recognized examples. In that study, the presence of certain bird species on a given island was associated with the absence of other species (and vice versa on other islands), resulting in a series of 'forbidden species combinations' or 'checkerboard distributions', which were posited to be the result of competitive exclusion (*Diamond, 1975*). This technique provided an important step forward in assessing the role of species interactions in field-based studies at the community level, but it has been frequently noted that analyses of species co-occurrence patterns need to include comparisons with patterns generated from communities assembled randomly to maximize inference (*Connor & Simberloff, 1979*; *Gotelli & Graves, 1996*).

Since the 1970s, species co-occurrence analyses have been used to assess the possibility of species interactions in a wide range of organisms, including both macro- and microorganisms (*Gotelli & McCabe, 2002*; *Horner-Devine et al., 2007*). Plant-associated fungal communities, which have diverse ecological roles in ecosystems (*Smith & Read, 2008*; *Rodriguez et al., 2009*), have shown a full range of co-occurrence patterns, including those consistent with both positive and antagonistic interactions (*Koide et al., 2005*; *Pan & May, 2009*; *Gorzelak, Hambleton & Massicotte, 2012*; *Ovaskainen, Hottola & Siitonen, 2010*; *Pickles et al., 2012*; *Toju et al., 2013*). For ectomycorrhizal (ECM) fungi, the dominant microbial eukaryotes in many temperate and some tropical forest soils (*Smith & Read, 2008*), these analyses have consistently found evidence of less species co-occurrence than expected by chance (*Koide et al., 2005*; *Pickles et al., 2010*; *Pickles et al., 2012*). This suggests that competitive interactions may play a significant role in structuring the communities of this fungal guild (*Kennedy, 2010*). The initial studies of species co-occurrence patterns in ECM fungal communities looked only in forests dominated by conifer hosts, but a recent study in *Fagus sylvatica* forests in Europe also found evidence of significantly lower than expected co-occurrence patterns (*Wubet et al., 2012*). This latter result indicates that the predominance of antagonistic interactions in determining ECM fungal community structure may be a common, host-lineage independent phenomenon. However, other ecological and evolutionary factors aside from species interactions can also be responsible for non-random species co-occurrence patterns (*Gotelli & McCabe, 2002*; *Ovaskainen, Hottola & Siitonen, 2010*), so caution must be applied in inferring underlying mechanisms.

In this study, we focused on assessing the community co-occurrence distributions of ECM fungi associated with the host genus *Alnus*. Unlike other ECM host genera with large geographical distributions, the ECM fungal communities associated with *Alnus* trees have been consistently found to be both species poor and highly host specific (*Tedersoo et al., 2009*; *Kennedy & Hill, 2010*; *Kennedy et al., 2011*; *Bogar & Kennedy, 2013*; *Põlme et al., 2013*; *Roy et al., 2013*). The mechanisms driving this atypical structure have long been thought to be related to the co-presence of nitrogen-fixing *Frankia* bacteria, which can have strong biotic and abiotic effects on *Alnus*-associated ECM fungal communities (*Walker et al., 2014*). In particular, the high rates of nitrification present in *Alnus* forest soils (due

to the high inputs and decomposition of nitrogen-rich leaf litter) results in significantly higher nitrate and acidity levels than those present in most other ECM-dominated forest soils *Danière, Capellano & Moiroud, 1986*; *Miller, Koo & Molina, 1992*; *Martin, Posavatz & Myrold, 2003*; *Walker et al., 2014*. Elevated levels of both of these abiotic factors have been shown to inhibit the growth of many ECM fungi (*Hung & Trappe, 1983*; *Lilleskov et al., 2002*) and, using an experimental pure culture approach, *Huggins et al. (in press)* recently demonstrated that *Alnus*-associated ECM fungi have a greater ability to tolerate high nitrate and acidity conditions compared to non-*Alnus*-associated ECM fungi.

Given the ability of *Alnus*-associated ECM fungi to grow in conditions that are generally considered abiotically stressful, we hypothesized that ECM fungal species co-occurrence patterns in *Alnus* forests may differ from those present in forests dominated by other ECM hosts. Specifically, we speculated that competitive interactions would be less prevalent in this study system, based on the fact that many studies of vascular plants have shown that the nature of species interactions often changes from antagonistic to positive with increasing levels of abiotic stress (*Bertness & Callaway, 1994*; *Gómez-Aparicio et al., 2004*, but see *Michalet et al., 2006*). To examine this hypothesis, we examined the co-occurrence patterns of the ECM fungal communities present in three mono-dominant plots of *Alnus rubra* in the western United States. ECM fungal communities were sampled on root tips and in soil. For the latter, we used sand-filled mesh in-growth bags, which allow for efficient, well-replicated community sampling of fungal hyphae growing in soil (*Wallander et al., 2001*; *Branco, Bruns & Singleton, 2013*). To identify the ECM fungi present in the study, we used high throughput Illumina sequencing, which has been increasingly used to profile ECM fungal community composition (*McGuire et al., 2013*; *Smith & Peay, 2014*).

## MATERIALS & METHODS

### Study location

The study site was located on the eastern side of the Coast Range mountains in northwestern Oregon, U.S.A. (latitude: N 45.820 W 123.05376, elevation: 462 m). Temperatures at the site are moderate (mean annual temperature $= 8.7\,°C$, min $= -1.2\,°C$, max $= 23.8\,°C$), with significant precipitation between October and May followed by drier summer months (total $= 1742$ mm). The specific study location is part of a long-term research project examining the effects of different forest management practices on *A. rubra* growth (see the Hardwood Silvicultural Cooperative (HSC) website for details, http://www.cof.orst.edu/coops/hsc). The HSC site used, Scappoose (HSC 3209), was established in 1995. Prior to the implementation of the HSC work, the site was a second-growth coniferous forest, which was clear-cut and replanted with a series of monodominant *A. rubra* plots. *A. rubra* seedlings were planted from nursery stock (Brooks Tree Farm, Brooks, OR) during the beginning of their second year of growth. Seedling ECM status at the time of planting was not assessed (*Frankia* nodules were noted to be absent), but nursery fumigation practices indicate colonization was unlikely (A Bluhm, pers. comm., 2009).

Our experiment was conducted in three 1,600 m$^2$ plots at HSC 3209. The plots, which were located approximately 100 m apart, differed in initial *A. rubra* stem density

(Plot 2 = 628, Plot 4 = 1,557, and Plot 8 = 3,559 stems/ha), but had no other forest management practices applied. Despite the differences in stem density, *A. rubra* fine root density did not differ significantly among the three plots (Fig. S1). The understories in all three plots were colonized by arbuscular mycorrhizal plants (dominated by *Mahonia nervosa* and *Claytonia perfoliata*), with no other ECM hosts besides *A. rubra* present. Soils were classified as well-drained Tolamy loams (USDA Soil Survey, Columbia County, OR). Within each plot, we located a 9 × 9 m subplot and overlaid a 100 point grid, with each point being separated by 1 m. We chose this subplot size to avoid any dead stems in the canopy immediately above the sampling area, while at the same time maximizing the number of samples taken per subplot. At each point in Plot 4, which was sampled for ECM root tips, a 5 cm diameter × 10 cm deep soil core was taken on May 31, 2013. In Plots 2 and 8, which were sampled for ECM communities present in soil, a 5 × 5 cm mesh bag was buried at each point 5 cm below the soil surface. The bags were made of anti-static polyester fabric with 300 μm diameter pores. This pore size allowed fungal hyphae to grow into the bags, but prevented penetration of plant roots. We filled the bags with twice autoclaved #3 grade Monterey aquarium sand (Cemex, Marina, CA, USA). Aluminum tags on fluorescent string were added to facilitate bag recovery. The mesh bags at Plot 2 were buried on February 1, 2013 and at Plot 8 on February 22. They were left undisturbed in the soil until May 31, when all were harvested. After removal from the soil, we placed the mesh bags into individual plastic bags and then onto ice for transport back to the laboratory. Soil cores and bags were stored at 4 °C for <96 h before further processing.

## Molecular analyses

We processed the root tip samples by gently washing all roots away from the soil and removing all ECM colonized root tips from each core under a 10X dissecting scope (∼10–50 root tips/core). All roots from each core were extracted using individual MoBio PowerSoil kits (Hercules, CA, USA), following manufacturer's instructions for maximum DNA yields. For the mesh bags, we followed the protocol outlined in *Branco, Bruns & Singleton (2013)*, which provided a cheaper and quicker protocol compared to direct DNA extraction from the sand within the mesh bags. Briefly, each bag (including a negative control that was taken to the field, but not buried) was emptied into a sterile 50 ml centrifuge tube. We added 10 ml of sterile deionized water and vortexed each tube for two minutes, followed by a five minute settling period (hyphae have been previously observed to float to the water surface). We then transferred the top two ml top of water to a new 2 ml centrifuge tube and contents were pelleted via centrifugation. On the same day, we extracted total genomic DNA from the pellets using the Sigma REDExtract-N-Amp kit (Sigma-Aldrich, St, Louis, MO, USA) following manufacturer's instructions. Root tips and extracts were stored for one week at −20 °C prior to PCR amplification.

For the root tip samples, we combined equal quantity aliquots from all 97 DNA extractions (three cores contained no roots) into a single template for PCR. In contrast, we conducted individual PCR reactions for each mesh bag sample as well as extraction controls. We processed these two types of samples differently because we were primarily

interested in the spatial co-occurrence patterns in the soil hyphal ECM fungal communities and therefore only used the root tip samples to create a local sequence reference set of known *Alnus*-associated ECM taxa against which the mesh bag data could be compared. For all PCR reactions, we used the barcoded ITS1F and ITS2 primer set of *Smith & Peay (2014)*, with each sample run in triplicate and pooled to minimize heterogeneity. Successful PCR products were determined by gel electrophoresis and magnetically cleaned using the Agencourt AMPure XP kit (Beckman Coulter, Brea, CA, USA) according to manufacturer's instructions. Final product concentrations were quantified using a Qubit dsDNA HS Fluorometer (Life Technologies, Carlsbad, CA, USA). Root tip and bag samples were run at different sequencing facilities under the same general conditions. For the root tips, the single PCR product was run at the University of Minnesota Genomics Center using 250 bp paired-end sequencing on the MiSeq Illumina platform. For the bags, we pooled the 192 successfully amplified bag samples at equimolar concentration and ran them on the same platform at the Stanford Functional Genomics Facility using 250 bp paired-end sequencing on the MiSeq Illumina platform. A spike of 20% and 30% PhiX was added to the runs to achieve sufficient sample heterogeneity, respectively. Raw sequence data and associated metadata from both the root tip and bag samples were deposited at MG-RAST (http://metagenomics.anl.gov/) under project #1080.

## Bioinformatic analyses

We used the software packages QIIME (*Caporaso et al., 2010*) and MOTHUR (*Schloss et al., 2009*) to process the sample sequences. Raw sequences were demultiplexed, quality filtered using Phred = 20, trimmed to 178 base pairs, and ends were paired, followed by filtering out of sequences that had any ambiguous bases or a homopolymer run of 9 bp. Following the guidelines discussed in *Nguyen et al. (in press)*, we employed a multi-step operational taxonomic unit (OTU) picking strategy by first clustering with reference USEARCH (including de novo chimera checking) at 97% sequence similarity, followed by UCLUST at 97% sequence similarity. We used a 97% similarity threshold because it was the most commonly employed in community-level ECM fungal studies, although some lineages, including *Alnicola*, may have greater sequence similarity among species (*Tedersoo et al., 2009*; *Rochet et al., 2011*). To assess the validity of the 97% threshold for sequences based on only ITS1 versus the full ITS region (i.e., ITS1, 5.8S, and ITS2), we examined seven known *Alnus*-associated *Tomentella* taxa (i.e., those present in *Kennedy et al., 2011*) and found that that threshold resulted in the same number of OTUs in both cases (data not shown). The UNITE database (*Kõljalg et al., 2013*) was used in both chimera checking and OTU clustering, with singleton OTUs discarded to minimize the effects of artifactual sequences (*Tedersoo et al., 2010*). We assigned taxonomic data to each OTU with NCBI BLAST+ v2.2.29 (*Altschul et al., 1990*), using a custom fungal ITS database containing the curated UNITE SH database (v6) (http://unite.ut.ee/repository.php, *Kõljalg et al., 2013*) and more than 600 vouchered fungal specimens, including 46 representative sequences from *Alnus* forests at other HSC locations in Oregon (*Kennedy & Hill, 2010*) and Mexico (*Kennedy et al., 2011*). Since sequences that had low subject length:query length matches

were typically non-fungal, we further filtered out sequences with matches ≤90% to BLAST (i.e., at least 90% of the bases in the input sequence matches to another sequence in the database at some identity level).

Using the remaining sequence dataset, we rarefied all samples to 12946 sequences, which was the lowest number of sequences obtained across the 192 samples. Since there has recently been a question raised about the validity of rarefaction in next generation sequencing analyses (*McMurdie & Holmes, 2014*), we also analyzed the data without rarefaction. We obtained very similar results (Table S1), so present the data based on rarefied samples only. ECM OTUs within each sample were parsed out using a python script that searches for genera names from a list of 189 known ECM genera and their synonyms (*Branco, Bruns & Singleton, 2013*, appended from *Tedersoo, May & Smith, 2010*). While this script provides a strong general filter for sorting the data by fungal lifestyle, some taxa belonging to clades that are polyphyletic for the ECM habit (e.g., *Lyophyllum*, *Sebacinales*) as well as taxa with low matches to Genbank (e.g., Uncultured Fungus) can be of questionable trophic status. For each of these groups, we carefully checked both the sequence matches and placement of our OTUs within phylogenetic trees of the clades to determine whether these taxa were properly classified at ECM. The resulting sample x OTU matrix contained 190 ECM taxa represented by at least one sequence per sample (min = 1, median = 34, mean = 1,334, max = 209,187). We found that 15 of the 190 OTUs present were highly similar (>97% similar) to ECM fungi present in the dipterocarp rainforests of Malaysia, which were concurrently being studied in the Peay lab using the same next-generation sequencing approach (Fig. 1). Because these OTUs represented accidental contamination probably during library construction, they were eliminated from the final analyses. Although an additional 80 OTUs had >97% similarity to taxa found in the Borneo study, because their closest BLAST match was not from Borneo, we conservatively considered these taxa as having cosmopolitan distributions and included them in the final analyses. The final OTU × sample matrix, including taxonomic matches and representative of sequences for each OTU, can be found in Table S2.

## Statistical analyses

Taxon co-occurrence patterns of the ECM fungal communities present in bag samples were assessed using the program EcoSim (*Gotelli & Entsminger, 2009*), with presence-absence matrices for Plots 2 and 8 being analyzed separately. (The root data from Plot 4 could not be analyzed for sample-level co-occurrence due to the pooled sequencing approach for those samples). We utilized the C-score algorithm (*Stone & Roberts, 1990*), which compares the number of checkerboard units (i.e., 1,0 × 0,1) between all pairs of species in the observed matrix ($C_{observed}$) to that based in random permutations of the same matrix ($C_{expected}$, i.e., the null models). Since randomized permutations of a matrix can be achieved in multiple ways (see *Gotelli & Entsminger, 2009* for details), we analyzed our datasets using both the 'fixed-fixed' and 'fixed-equiprobable' options (which are recommended by the program guide and used in the previous ECM fungal co-occurrence analyses). In both options, the row (i.e., taxon) totals were fixed, so that the

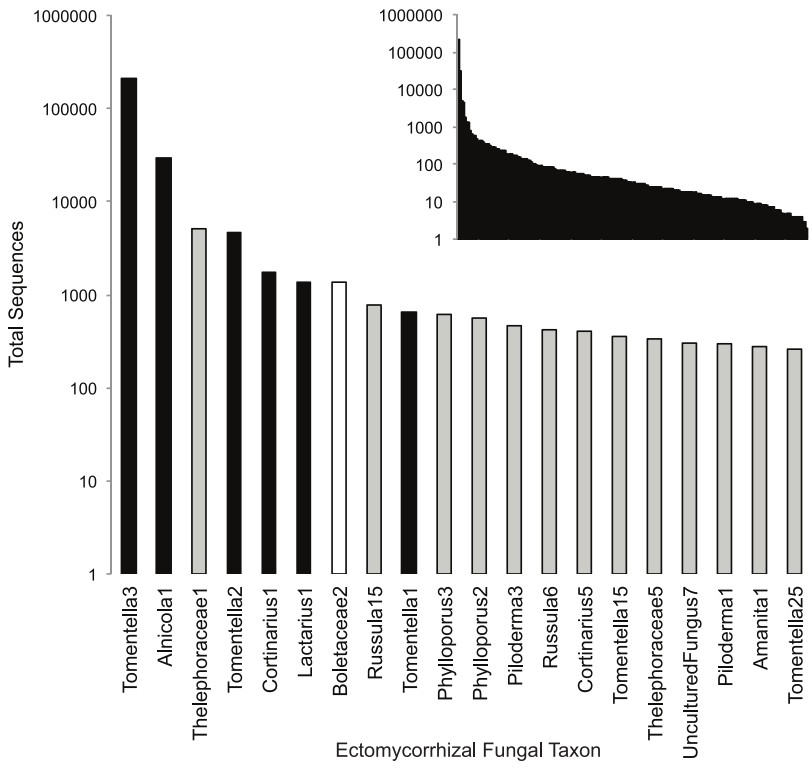

**Figure 1 Rank-abundance plot of all 190 (inset) and top 20 ectomycorrhizal (ECM) fungal taxa sampled in this study.** The top 20 ECM fungal taxa are color coded by whether they are known to be associated with *Alnus* hosts (black), of unknown host origin (grey), or laboratory contaminants (white).

total abundances of each taxon in the observed and null matrices were identical. In the 'fixed-equiprobable' option, however, the column (i.e., sample) totals in the null matrices were no longer equivalent to those in the observed matrix. Instead, all samples in the null matrices had an equal probability of being colonized by any of the taxa in the observed matrix, which effectively eliminates differences in taxon richness among samples.

Of the ECM fungal taxa present in the final root tip and bag datasets, over 90% (167/175) belonged to species never previously encountered with *Alnus* (Table S2, *Alnus*Match = No). Unlike other ECM host systems with large geographic ranges, the ECM fungal community associated with *Alnus* hosts is remarkably well characterized at local (*Tedersoo et al., 2009*; *Kennedy & Hill, 2010*; *Walker et al., 2014*), regional (*Kennedy et al., 2011*; *Roy et al., 2013*), and global scales (*Põlme et al., 2013*). As such, it is highly likely the majority of the novel OTUs encountered were not part of the active ECM community in our plots, but rather present simply either as spores or additional lab contaminants. To account for this issue, we divided our checkerboard analyses into five different input matrices for the bag dataset (Plots 2 and 8). The first matrix included all 175 ECM fungal taxa (referred to as "All"). The second matrix included the 16 taxa that had >97% similarity matches to ECM samples from *Alnus* forests (referred to as *Alnus*). The third matrix included only the 8 taxa that were encountered on ECM root tips in Plot 4 (referred to as *Alnus*RootOnly). To assess the robustness of the results generated

using the larger *Alnus* matrix, the fourth matrix excluded the three most frequent and abundant species (*Tomentella*3, *Alnicola*1, *Tomentella*2) (referred to as *Alnus*MinusTop3). Finally, the fifth matrix included just the 10 taxa in the genus *Tomentella* (from the larger *Alnus* matrix) to look for evidence of species interactions among this subset of closely related taxa (referred to as *AlnusTomentella*Only). For all of the aforementioned C-score analyses, taxa present in less than 5 bag samples were removed, as low frequency taxa are generally considered non-informative (*Koide et al., 2005*). The observed input matrices were compared to 5000 null matrices. Significant differences between the observed matrix C-score and that of the null matrices were determined along with standardized effect sizes (SES). Observed C-scores significantly higher than those generated from the null matrices are consistent with a community being structured by competitive interactions, whereas $C_{observed}$ significantly lower than the $C_{expected}$ is consistent with positive interactions.

To further assess the degree of association among known *Alnus* ECM fungal taxa, we also used an abundance-based approach (as opposed to the co-occurrence analyses, which are based on binary presence/absence data). Specifically, we calculated the pair-wise Spearman rank correlation coefficients among all pairs of the 16 *Alnus*-associated taxa using the *cor* function in R (*R Core Team, 2013*). Coefficients >0.30 were tested for significance with the *cor.test* function. To account for multiple tests ($n = 13$), we used a Bonferroni-corrected *P* value of 0.003. With the same data set, we also tested for the presence of spatial autocorrelation using the *mgram* function in the ECODIST package in R. We first converted the sequence abundance datasets in both Plots 2 and 8 into dissimilarity matrices using the Bray-Curtis Index and then compared those to a Euclidean distance matrix of sampling points for each plot. For the Mantel correlogram tests, we used the *n.class = 0* option, which uses Sturge's equation to determine the appropriate number of distance classes.

## RESULTS

We found 175 total ECM fungal taxa in the study (Table S2); 16 of which matched closely to known *Alnus*-associated ECM fungi. In the mesh bags, *Alnus*-associated ECM fungal taxa represented six of the ten most abundant OTUs present, including the dominant ECM fungal taxon, *Tomentella*3, which was present in all the bag samples in both plots and had sequence abundances nearly ten-fold higher than any other taxon (Figs. 2A and 2B). Two other *Alnus*-associated fungal taxa, *Alnicola*1 and *Tomentella*2, were also present in all samples, whereas the remaining *Alnus*-associated ECM fungal taxa had frequencies varying from 2 to 96% (Plot 2 mean = 25%, Plot 8 mean = 31%) and lower sequence abundances. Eight of the 16 *Alnus*-associated ECM fungal taxa were present on both roots and in the bags, with abundances that were very similar (Fig. 1A). Of the eight ECM fungal taxa found on root tips, all were previously encountered on *A. rubra* root tips at other sites in Oregon, while the eight fungal taxa found exclusively in bags had not been previously documented (*Kennedy & Hill, 2010*).

ECM fungal taxon co-occurrence patterns were largely consistent between plots, but different between null models. Of the ten tests (i.e., 5 matrix types × 2 plots) using the

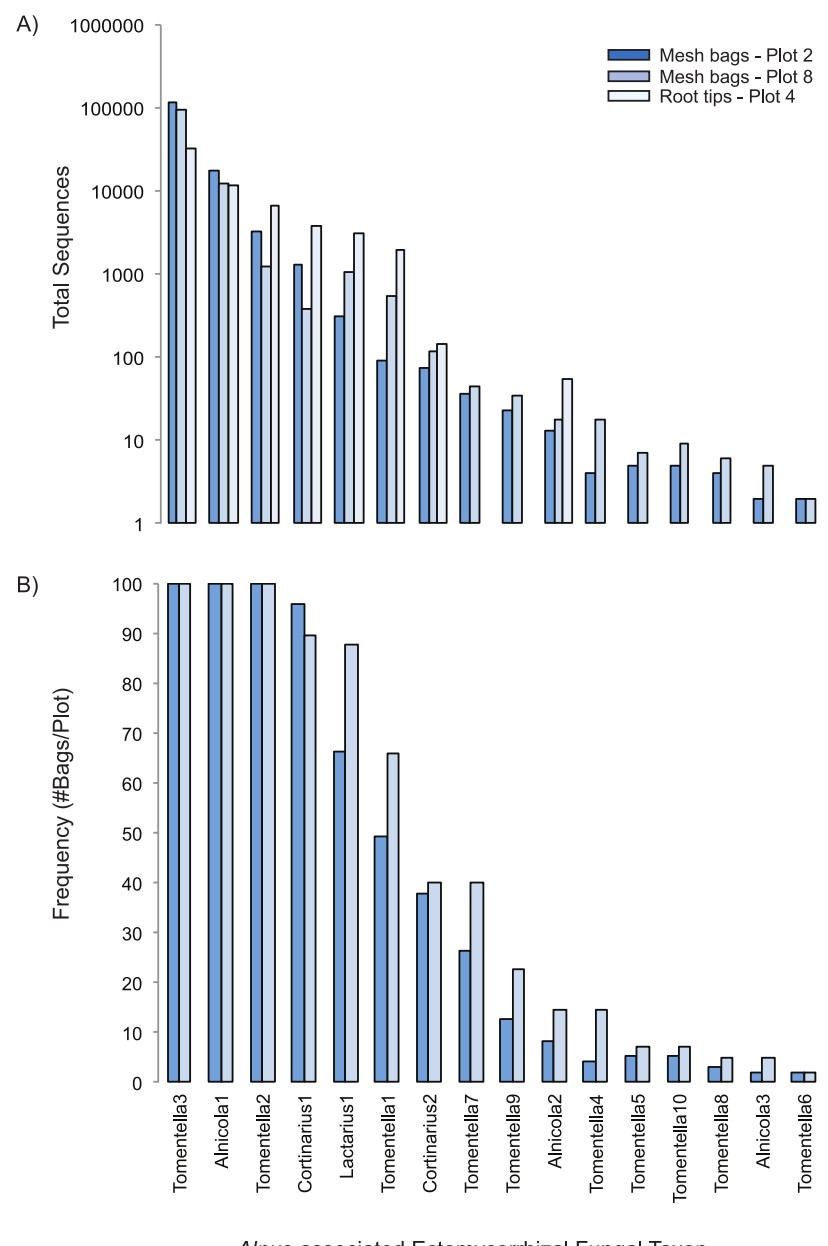

**Figure 2** Rank-abundance (A) and rank-frequency (B) plots of *Alnus*-associated ectomycorrhizal fungal taxa sampled in mesh bags and root tips.

'fixed-fixed' permutation option, nine indicated that the observed ECM fungal community did not differ significantly from random assembly (Table 1). In one case, Plot 2 All, the observed ECM fungal community had significantly more co-occurrence than expected by chance. In contrast, in the ten tests using the 'fixed-equiprobable' permutation option, three indicated that the observed ECM fungal community did not differ significantly from random assembly, while seven found that the observed ECM fungal community had significantly more co-occurrence than expected by chance. Results remained the same for

**Table 1** **C-score taxon occurrence analyses of ECM fungal communities in Plots 2 and 8.** See methods for details about datasets and null matrix type definitions.

| Dataset | Plot | Null matrix type | C observed | C expected | P value | SES |
|---|---|---|---|---|---|---|
| All | 2 | Fixed–Fixed | 173.2 | 173.8 | 0.00 | −3.35 |
| | | Fixed-Equiprobable | | 188.1 | 0.00 | −21.2 |
| All | 8 | Fixed–Fixed | 164.4 | 164.7 | 0.73 | −0.75 |
| | | Fixed-Equiprobable | | 172.1 | 0.00 | −11.8 |
| *Alnus* | 2 | Fixed–Fixed | 93.5 | 92.5 | 0.21 | 0.76 |
| | | Fixed-Equiprobable | | 106.7 | 0.04 | −1.75 |
| *Alnus* | 8 | Fixed–Fixed | 103.1 | 103.2 | 0.47 | −0.07 |
| | | Fixed-Equiprobable | | 114.5 | 0.04 | −1.82 |
| *Alnus*RootOnly | 2 | Fixed–Fixed | 77.4 | 76.7 | 0.74 | 0.54 |
| | | Fixed-Equiprobable | | 82.5 | 0.27 | −0.59 |
| *Alnus*RootOnly | 8 | Fixed–Fixed | 61.2 | 61.6 | 0.45 | −0.26 |
| | | Fixed-Equiprobable | | 78.4 | 0.13 | −1.15 |
| *Alnus*MinusTop3 | 2 | Fixed–Fixed | 200.3 | 198.4 | 0.27 | 0.59 |
| | | Fixed-Equiprobable | | 228.25 | 0.04 | −1.72 |
| *Alnus*MinusTop3 | 8 | Fixed–Fixed | 178.7 | 179.3 | 0.47 | −0.20 |
| | | Fixed-Equiprobable | | 198.6 | 0.03 | −1.88 |
| *Alnus*TomentellaOnly | 2 | Fixed–Fixed | 61.6 | 62.6 | 0.77 | −0.55 |
| | | Fixed-Equiprobable | | 88.7 | 0.02 | −1.99 |
| *Alnus*TomentellaOnly | 8 | Fixed–Fixed | 108.3 | 107.6 | 0.64 | 0.31 |
| | | Fixed-Equiprobable | | 109.1 | 0.47 | −0.09 |

**Notes.**
SES, Standardized Effect Size.

*Alnus* ECM fungal communities whether the top three taxa were removed or not. The *Alnus* and *Alnus*RootOnly analyses did differ under the 'fixed-equiprobable' option, with the former showing greater than expected co-occurrence and the latter having a pattern no different than one based on random assembly. Additionally, in the *AlnusTomentella*Only analysis, the ECM fungal community showed greater than expected co-occurrence in Plot 2 but not in Plot 8. In all of these cases, significant antagonistic patterns were not observed.

Spearman rank analyses revealed that pair-wise sequence abundances of some of the 16 *Alnus* ECM fungal taxa were significantly positively correlated (Table 2). The specific significant combinations varied between plots, with only one taxon pair (*Alnicola*1 & *Tomentella*9) showing significant positive correlations in both plots. Although a number of pair-wise correlations had negative values (suggesting negative rather than positive interactions), none of them were significant, even when considered at a *P* value of 0.05. In addition, the Mantel correlogram analyses found no clear evidence of spatial autocorrelation in the *Alnus*-associated ECM fungal communities. In Plot 2, there was no significant autocorrelation at any distance, while in Plot 8 there was a single significant positive correlation between samples located 1–2 m apart (Figs. S2 and S3).

**Table 2 Spearman rank correlation coefficient matrices for ECM fungal communities in Plots 2 and 8.** Significant correlations are indicated in bold. Numbers over the columns of both matrices correspond to the number of the ECM fungal taxon identified in the first row.

| Plot 2 | 1 | 2 | 3 | 4 | 5 | 6 | 7 | 8 | 9 | 10 | 11 | 12 | 13 | 14 | 15 | 16 |
|---|---|---|---|---|---|---|---|---|---|---|---|---|---|---|---|---|
| 1. Tomentella3 | 1.00 | | | | | | | | | | | | | | | |
| 2. Alnicola1 | 0.00 | 1.00 | | | | | | | | | | | | | | |
| 3. Tomentella2 | −0.02 | 0.00 | 1.00 | | | | | | | | | | | | | |
| 4. Cortinarius1 | −0.05 | −0.07 | 0.01 | 1.00 | | | | | | | | | | | | |
| 5. Lactarius1 | −0.07 | −0.07 | −0.02 | 0.28 | 1.00 | | | | | | | | | | | |
| 6. Tomentella1 | −0.08 | 0.09 | 0.00 | −0.13 | −0.06 | 1.00 | | | | | | | | | | |
| 7. Cortinarius2 | −0.13 | 0.11 | 0.01 | −0.05 | 0.00 | 0.26 | 1.00 | | | | | | | | | |
| 8. Tomentella7 | 0.16 | −0.08 | **0.65** | −0.03 | 0.08 | −0.04 | 0.10 | 1.00 | | | | | | | | |
| 9. Tomentella9 | 0.11 | **0.48** | −0.06 | −0.05 | 0.00 | 0.12 | 0.06 | −0.02 | 1.00 | | | | | | | |
| 10. Alnicola2 | 0.07 | **0.42** | 0.07 | −0.05 | −0.04 | −0.02 | −0.10 | 0.02 | 0.09 | 1.00 | | | | | | |
| 11. Tomentella4 | −0.08 | −0.04 | **0.40** | −0.01 | 0.18 | 0.01 | 0.00 | 0.26 | 0.00 | 0.04 | 1.00 | | | | | |
| 12. Tomentella5 | 0.15 | −0.07 | −0.04 | −0.06 | −0.04 | −0.03 | 0.00 | −0.10 | −0.01 | −0.06 | −0.05 | 1.00 | | | | |
| 13. Tomentella10 | 0.06 | 0.01 | −0.02 | −0.03 | 0.00 | 0.08 | −0.10 | 0.01 | −0.05 | −0.06 | −0.05 | −0.06 | 1.00 | | | |
| 14. Tomentella8 | −0.07 | −0.03 | **0.40** | 0.01 | −0.03 | −0.06 | 0.03 | 0.25 | −0.04 | 0.04 | **0.60** | −0.04 | −0.04 | 1.00 | | |
| 15. Alnicola3 | **0.39** | −0.06 | 0.27 | −0.04 | −0.04 | −0.11 | 0.07 | **0.50** | 0.03 | −0.04 | −0.03 | 0.29 | −0.03 | −0.02 | 1.00 | |
| 16. Tomentella6 | **0.37** | −0.06 | −0.03 | −0.03 | −0.02 | −0.11 | −0.08 | 0.02 | 0.03 | −0.04 | −0.03 | −0.03 | −0.03 | −0.02 | −0.02 | 1.00 |
| Plot8 | 1 | 2 | 3 | 4 | 5 | 6 | 7 | 8 | 9 | 10 | 11 | 12 | 13 | 14 | 15 | 16 |
| 1. Tomentella3 | 1.00 | | | | | | | | | | | | | | | |
| 2. Alnicola1 | 0.13 | 1.00 | | | | | | | | | | | | | | |
| 3. Tomentella2 | −0.17 | −0.12 | 1.00 | | | | | | | | | | | | | |
| 4. Cortinarius1 | 0.03 | −0.03 | 0.26 | 1.00 | | | | | | | | | | | | |
| 5. Lactarius1 | −0.16 | −0.09 | 0.14 | 0.14 | 1.00 | | | | | | | | | | | |
| 6. Tomentella1 | −0.14 | 0.03 | 0.04 | 0.04 | **0.46** | 1.00 | | | | | | | | | | |
| 7. Cortinarius2 | 0.06 | **0.45** | −0.04 | −0.02 | −0.05 | −0.06 | 1.00 | | | | | | | | | |
| 8. Tomentella7 | −0.09 | −0.06 | −0.06 | 0.14 | 0.11 | 0.02 | 0.05 | 1.00 | | | | | | | | |
| 9. Tomentella9 | −0.12 | **0.47** | 0.02 | −0.01 | 0.10 | 0.16 | 0.28 | 0.06 | 1.00 | | | | | | | |
| 10. Alnicola2 | −0.06 | 0.15 | 0.02 | 0.12 | −0.04 | 0.03 | 0.08 | 0.13 | 0.07 | 1.00 | | | | | | |
| 11. Tomentella4 | −0.05 | −0.07 | 0.15 | 0.12 | 0.14 | −0.02 | −0.09 | 0.18 | 0.02 | 0.07 | 1.00 | | | | | |
| 12. Tomentella5 | −0.05 | 0.05 | 0.07 | −0.07 | 0.14 | −0.01 | −0.08 | −0.08 | 0.08 | −0.10 | 0.22 | 1.00 | | | | |
| 13. Tomentella10 | 0.08 | −0.04 | −0.05 | 0.02 | 0.02 | 0.03 | 0.13 | −0.05 | −0.12 | 0.12 | −0.04 | −0.07 | 1.00 | | | |
| 14. Tomentella8 | 0.16 | −0.07 | 0.01 | 0.06 | −0.04 | 0.00 | −0.06 | 0.06 | −0.06 | 0.26 | 0.14 | −0.06 | −0.05 | 1.00 | | |
| 15. Alnicola3 | 0.28 | 0.24 | −0.10 | −0.11 | 0.00 | −0.06 | 0.21 | −0.10 | 0.14 | −0.08 | −0.09 | −0.07 | 0.07 | −0.05 | 1.00 | |
| 16. Tomentella6 | −0.02 | −0.04 | 0.18 | −0.04 | −0.03 | −0.04 | 0.06 | −0.10 | **0.33** | −0.05 | −0.06 | −0.04 | −0.04 | −0.03 | −0.03 | 1.00 |

## DISCUSSION

We found that the ECM fungal communities in *A. rubra* forests displayed a different pattern of taxon co-occurrence compared to those seen for other ECM fungi. Unlike the consistent previous findings of less co-occurrence among species than expected by chance (*Koide et al., 2005*; *Pickles et al., 2012*; *Wubet et al., 2012*), we observed no evidence of spatial patterns consistent with interspecific competition in *Alnus*-associated ECM fungal communities. In contrast, we consistently found co-occurrence patterns that

**Peer**J

were either no different from random assembly or consistent with positive interactions. Although we did not measure soil nitrate and acidity conditions in this study (see *Martin, Posavatz & Myrold (2003)* and *Walker et al. (2014)* for values from comparable age *A. rubra* forests at other sites in Oregon), *Alnus* soils are consistently characterized by abiotic conditions are generally considered stressful to ECM fungi. The results we obtained are thus consistent with the 'stress gradient hypothesis', which posits that species interactions shift from negative to positive as environmental conditions become harsher (*Bertness & Callaway, 1994*). Although we emphasize that the patterns we found in this study are based solely on correlative inference, there is some experimental evidence that may support the stress gradient hypothesis for ECM fungal community dynamics. *Koide et al. (2005)* found a shift from significant negative co-occurrence patterns in their control plots to non-significant co-occurrence patterns in plots where either tannins or nitrogen were added experimentally. While they did not explicitly analyze these manipulations in terms of stress, both increased tannin and nitrogen levels have been shown to inhibit the growth of multiple ECM fungal taxa (*Koide et al., 1998*; *Cox et al., 2010*). The direction of the response in the *Koide et al. (2005)* study is consistent with greater abiotic stress resulting in a decrease in antagonistic ECM fungal interactions. At the same time, it is plausible that resource limitation was eliminated with the addition of nitrogen, which could have allowed for greater spatial co-existence among ECM fungi. Since the *Alnus* system has naturally higher nitrogen availability than most ECM forests due to the co-presence of nitrogen-fixing *Frankia* bacteria, it is also possible that greater resource abundance could drive the co-occurrence patterns we observed. Given the fact that the pattern could be explained by either increasing stress or resource availability, additional studies are needed to distinguish among these explanations. One promising approach would be to examine the taxon co-occurrence patterns in younger and older *Alnus* forests, since soil nitrate and acidity concentrations increase in these forests over time (*Danière, Capellano & Moiroud, 1986*; *Martin, Posavatz & Myrold, 2003*). If the stress gradient hypothesis were the most plausible explanation, then we would expect to see competitive and facilitative interactions to be dominant, respectively.

The presence of co-occurrence patterns consistent with significant negative species interactions was also missing in our analysis of more closely related ECM fungal taxa. For the ten *Alnus*-associated members of the genus *Tomentella*, co-occurrence patterns either did not differ significantly from random assembly or reflected an effect of positive interactions. Like the larger community analyses, this result also differs from previous experimental studies, where strong antagonistic interactions among closely related ECM fungal taxa have been observed (*Kennedy, 2010*). In a similarly designed study that also assessed ECM fungi with taxon co-occurrence analyses, *Pickles et al. (2012)* found patterns consistent with strong interspecific competition among a suite of *Cortinarius* species in a Scottish *Pinus sylvestris* forest. Although it has long been assumed that competition may be stronger in more closely related species due to greater overlap in resource utilization, a meta-analysis by *Cahill et al. (2008)* found little consistent evidence to support this supposition. *Mayfield & Levine (2010)* further questioned the validity of phylogenetic

**Peer**J

relatedness as a good proxy for competitive strength by showing that in certain abiotic environments competition may actually select for more closely related taxa than expected by chance (i.e., phylogenetic clustering). The *Alnus* ECM system is particularly interesting in this respect because while the fungal communities associated with *Alnus* hosts are both species poor and highly host specific, they include taxa from a number of distantly related lineages (*Rochet et al., 2011*). Although explanations for this higher-level phylogenetic patterning are still lacking, our current results suggest that competitive processes among both closely and more distantly related taxa are not a key factor generating the atypical structure of *Alnus* ECM fungal communities.

Some positive spatial associations have been observed in other studies of ECM fungal communities (*Agerer, Grote & Raidl, 2002*; *Koide et al., 2005*; *Pickles et al., 2012*), and have been suggested to be due to complementary resource acquisition abilities of among individual taxa (*Jones et al., 2010*). We speculate that in *Alnus* forests positive associations among ECM fungi could also reflect possible amelioration of local abiotic conditions. *Huggins et al. (in press)* found that *Alnus*-associated ECM fungi could more effectively buffer changes in local pH environments than non-*Alnus* ECM fungi, which may be key to persistence in the high acidity soils present in *Alnus* forests. While the exact buffering mechanism is not yet known, if it involves the release of molecules into the external environment, growing directly adjacent to another ECM fungus may result in greater buffering of local pH conditions than when growing in isolation. We believe it is important to note, however, that the patterning of positive associations were patchy and not consistent between plots, so it is hard to determine if local pH buffering is actually significant without local measurements of pH for each sample. Furthermore, sequence abundance of individual taxa has been shown not to correlate linearly with initial fungal tissue or DNA abundance in other studies using NGS techniques (*Amend, Seifert & Bruns, 2010*; *Nguyen et al., in press*), so caution must be applied in using sequence abundance as an accurate ecological proxy.

Like the co-occurrence and correlation-based patterns, we found that spatial auto-correlation patterns observed in *Alnus* ECM fungal communities were also anomalous relative to other studies. The specific distance of spatial autocorrelation appears to vary among systems, but there is typically strong spatial autocorrelation among community samples located less than 5 m apart (e.g., *Lilleskov et al., 2004*; *Bahram et al., 2013*). While the spatial extent of our study was very limited (the most distant samples within plots were only ∼12 m apart), the absence of spatial signal was not surprising, based on previous studies of *Alnus* ECM fungal communities. Both *Pritsch et al. (2010)* and *Kennedy et al. (2011)* found individual *Alnus* ECM fungal taxa that were almost identical genetically (at least in the ITS region) in forests located thousands of kilometers apart and, in a global scale analysis, *Põlme et al. (2013)* found many *Alnus* ECM OTUs were distributed across geographically distant samples. Theoretically, the absence of dispersal limitation should make the detection of non-random distribution patterns more likely if biotic interactions (either negative or positive) are strong determinants of community structure. The classic work of *Diamond (1975)* is a good example, as the bird populations

studied across the New Guinea archipelago were not dispersal limited, yet exhibited many checkerboard distribution patterns. As such, we do not think the atypical nature of the taxon co-occurrence patterns in *Alnus* ECM fungal communities that we observed were driven by the also atypical spatial correlation patterns.

As the results observed in this study differed in multiple ways from those found previously, we had some concern they were caused by an artifact of our identification or sampling methodology. Unlike previous examinations of taxon co-occurrence for ECM fungi, we used next-generation sequencing (NGS) to identify the communities present. NGS methods provide much greater sequencing depth per sample (*Smith & Peay, 2014*), which may have allowed us to more effectively document the ECM fungal communities present in each sample compared to previous studies. We found that the three most abundant *Alnus*-associated ECM fungi were present in every bag sample in both plots, which has not been observed in other systems. Although the presence of spatially ubiquitous taxa will result in a lower total number of checkerboard units observed (as 1,0 is possible but not 0,1), it has the same effect on both the observed and null matrices and therefore should not bias statistical comparisons of $C_{observed}$ versus $C_{expected}$. We checked this by eliminating the three ECM fungal taxa present in every sample and found functionally identical results to those when those taxa were included (Table 2). A second difference between this and related studies was the sampling of ECM fungal hyphal communities in mesh bags. Previous studies assessing co-occurrence patterns have largely focused on ECM root tips, but *Koide et al. (2005)* found very similar taxon co-occurrence patterns for root-tip and soil-based analyses of ECM fungal communities in the same *Pinus resinosa* forest. Based on that result, and the fact that the sequence abundances of all the ECM fungi present on *A. rubra* root tips and the mesh bags showed highly similar patterns (Table S2), we do not believe assessing ECM hyphal communities was the source of our incongruous results either (however, in hindsight, a better experimental design would have been to sample the mesh bags and the ECM root tips directly around them within each plot). A third difference is the restricted taxonomic richness of *Alnus* ECM fungal communities. This explanation, however, also seems unwarranted, as *Pickles et al. (2012)* showed highly significant negative co-occurrence patterns in matrices of equivalent sizes. Finally, it is also possible that variation in soil nutrient availability could drive *Alnus* ECM fungal community structure and, because it was relatively homogenous in our small-sized plots, the resulting taxon distribution patterns were largely random. While we reiterate that we did not directly measure soil nutrient availability in this study, other studies of *Alnus* ECM fungi have shown some significant correlations between community structure and soil organic matter and nutrients such as K and Ca (*Becerra et al., 2005*; *Tedersoo et al., 2009*; *Roy et al., 2013*, *Põlme et al., 2013*; see *Richard (1968)* for a possible mechanism). In those studies, however, the percent of variance explained by soil nutrients was generally low, so we believe it is unlikely that variation in resource availability was the primary determinant of the distribution patterns observed. We recognize that additional differences likely exist, but feel confident that the co-occurrence results we observed are ecologically accurate and not generated by methodological or sampling artifact.

NGS techniques clearly represent a powerful and efficient way to assess the richness and dynamics of fungal communities (*Smith & Peay, 2014*), but we found that additional data quality control analyses beyond the standard sequence quality thresholds and chimera checking were needed to properly characterize ECM fungal community composition. Specifically, we found that a relatively high number of ECM fungal taxa present appeared to be the result of PCR contamination. The PCR reactions of our extraction and PCR controls produced no bands indicating positive product, but the sensitivity of NGS techniques and the Illumina platform in particular makes the amplification of single DNA molecules highly probable (*Tedersoo et al., 2010*; *Peay, Baraloto & Fine, 2013*). Fortunately, the atypical and well-described nature of *Alnus* ECM fungal communities made it relatively easy to identify the most obvious non-*Alnus* associated fungal taxa and remove them prior to the final analyses. For taxa that belonged to ECM fungal lineages known to associate with *Alnus* hosts but which had not been previously documented, it was more difficult to determine their status (i.e., whether they represented PCR contaminants, were present in *A. rubra* soils as spores, or actually colonizing *A. rubra* root tips). In particular, the status of Thelephoraceae1, which had the third highest sequence abundance in the full dataset, was interesting because the closest BLAST match to Thelephoraceae1 was an ECM fungal root tip sample from *Betula occidentalis* in British Columbia, Canada. *Bogar & Kennedy (2013)* found that ECM fungal communities present on *Alnus* and *Betula* hosts can overlap, so it is possible this taxon was overlooked in previous surveys of *Alnus* ECM fungal communities that used less sensitive methods. However, the absence of this taxon from any the root tip samples in Plot 4 suggests that it was most likely present simply as spores rather than an active member of the *Alnus*-associated ECM fungal community. Despite the unclear status of this taxon as well as many others with lower abundance, the co-occurrence patterns showed the same general results whether taxa of unknown status were included or not, suggesting the overall results were robust. In less well-characterized ECM fungal and other microbial systems, however, the potential for inclusion of spurious taxa is sufficiently high that we strongly recommend the sequencing of negative extraction and PCR controls to help try to account for any lab-based contamination (*Nguyen et al., in press*).

Taken together, our results suggest that while many ECM fungal communities appear to be strongly affected by competitive interactions, those present in *Alnus* forests are not. Although the reasons for this difference are not fully resolved in this study, the possibility of greater abiotic stress changing the way in which species interact in *Alnus* forests is likely an important factor. The application of ecological theories such as the stress gradient hypothesis to better understand the factors driving ECM fungal community structure has grown rapidly in recent years (*Peay, Kennedy & Bruns, 2008*; *Koide, Fernandez & Malcolm, 2014*) and new technologies such as next generation sequencing continue to make the study of ECM fungi increasingly tractable for ecologists. While we welcome this synergy, we stress the importance of a solid foundation in fungal biology as well as a critical awareness of the limitations of molecular-based identification techniques to successfully integrate ECM fungi into the ecological mainstream.

## ACKNOWLEDGEMENTS

We thank A Bluhm and D Hibbs for assistance using the HSC study location, L Bogar, V Engebretson, J Huggins, P King for assistance with experiment implementation and harvest, J Walker for assistance with DNA extractions, D Smith for assistance with NGS processing, and members of the Peay Lab, C Fernandez, R Koide, M Gardes and one anonymous reviewer for critical comments on a previous version of this manuscript.

### Funding

Support for this work came from NSF DEB Grant #1030275 to Peter Kennedy. The funders had no role in study design, data collection and analysis, decision to publish, or preparation of the manuscript.

### Grant Disclosures

The following grant information was disclosed by the authors:
NSF DEB: #1030275.

### Competing Interests

The authors declare there are no competing interests.

### Author Contributions

- Peter Kennedy conceived and designed the experiments, analyzed the data, wrote the paper, prepared figures and/or tables.
- Nhu Nguyen analyzed the data, contributed reagents/materials/analysis tools, reviewed drafts of the paper.
- Hannah Cohen performed the experiments, reviewed drafts of the paper.
- Kabir Peay contributed reagents/materials/analysis tools, reviewed drafts of the paper.

### Field Study Permissions

The following information was supplied relating to field study approvals (i.e., approving body and any reference numbers):

No permit was needed as the study was conducted on private land.

### DNA Deposition

The following information was supplied regarding the deposition of DNA sequences:

We have provided access to the raw sequence reads with the MG-RAST (http://metagenomics.anl.gov/) under project #1080.

### Supplemental Information

Supplemental information for this article can be found online at http://dx.doi.org/10.7717/peerj.686#supplemental-information.

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
