# Peer review of "Missing checkerboards? An absence of competitive signal in Alnus-associated ectomycorrhizal fungal communities"

_PeerJ, doi:10.7717/peerj.686_

## Round 0.1 · original submission · Minor Revisions

Dear Peter,

Referees and I are very pleased with your revisions and responses, and am thus happy to recommend publication of your revised manuscript. Referee #1 however suggested a few minor revisions (see below) that you should take into account in your final version. Referee #2 found one reference that needs to be edited..

None of the comments should be too difficult to deal with. If you would upload the revised version in the coming days, I will check it over and let you know my final decision.  I am hopeful it will not require further peer review.

I look forward to receiving your revised version.

Cheers - Francis

Reviewer 1 ·

Basic reporting

No comments.

Experimental design

No comments.

Validity of the findings

It is good to see that the authors decided not to "try to hide the issues with our data with some clever bioinformatics binning or forceful 'trust-me' arguments". Unfortunately, that statement makes one wonder how often this is being done.

The title merits a question mark because the results are doubly atypical - regarding co-occurrence and autocorrelation. Exceptional results call for exceptionally strong evidence, which this article does not present due to a variety of methodological uncertainties and caveats that the authors acknowledge.

Additional comments

Corrections:
Abstract: ….and DNA sequence abundance…
….from root tips or fungal in-growth bags in three A. rubra plantation plots at a site in Oregon, USA, and identified…
Introduction: Lilleskov et al. 2004 merits being mentioned here.
98. …were colonized by arbuscular…
143. root tips
178. rarefaction
Discussion: Mention that sampling roots from the same plot as where the bags were buried would have been preferable.
Figure 1. What is the evidence that UnculturedFungus7 is ECM?
Table S2. The naming of all taxa needs to be checked and corrected when necessary. For example, an OTU 97% similar to Amphinema is named UnculturedFungus4, an OTU 96.5% similar to Piloderma is named Herpothichiellaceae1 (sic), etc.

·

Basic reporting

I found one reference that still needs to be corrected. The correct title of the paper of Walker et al. 2014 is "....in a global tripartite symbiosis" (not in a tri-partite symbiosis" as it is written), please change it.

Experimental design

no comments

Validity of the findings

no comments

Additional comments

The authors have improved significantly the quality of the manuscript. All my major concerns have been addressed.

---

## Round 0.2 · accepted · Accept

Dear Peter,

Thanks for the revised version of your ms. I am happy to tell you that it is provisionally accepted for publication in PeerJ. I have forwarded my recommendation for publication to the PeerJ staff and they will be in touch shortly to move forward with publication.

BTW, I've spotted a typo in your last ms. version: 'by arbsucular mycorrhizal plants' should read 'by arbuscular mycorrhizal plants'.